# Information–Theoretic Aspects of Location Parameter Estimation under Skew–Normal Settings

**DOI:** 10.3390/e24030399

**Published:** 2022-03-13

**Authors:** Javier E. Contreras-Reyes

**Affiliations:** Instituto de Estadística, Facultad de Ciencias, Universidad de Valparaíso, Valparaíso 2360102, Chile; jecontrr@uc.cl

**Keywords:** skew–normal distribution, location parameter, skewness, differential entropy, Fisher information, Cramér–Rao bound, convexity

## Abstract

In several applications, the assumption of normality is often violated in data with some level of skewness, so skewness affects the mean’s estimation. The class of skew–normal distributions is considered, given their flexibility for modeling data with asymmetry parameter. In this paper, we considered two location parameter (μ) estimation methods in the skew–normal setting, where the coefficient of variation and the skewness parameter are known. Specifically, the least square estimator (LSE) and the best unbiased estimator (BUE) for μ are considered. The properties for BUE (which dominates LSE) using classic theorems of information theory are explored, which provides a way to measure the uncertainty of location parameter estimations. Specifically, inequalities based on convexity property enable obtaining lower and upper bounds for differential entropy and Fisher information. Some simulations illustrate the behavior of differential entropy and Fisher information bounds.

## 1. Introduction

A typical problem in statistical inference is estimating the parameters from a data sample [1], especially if the data have some level of skewness. Therefore, the estimation of these parameters is affected by asymmetry. Recent research addressed data asymmetry with the class of skew–normal distributions, given their flexibility for modeling data with the skewness (asymmetry/symmetry) parameter [2]. In particular, Ref. [3] focused on estimating location parameter (μ), assuming that the coefficient of variation and skewness parameter are known. Specifically, they presented the least square estimator (LSE) and the best unbiased estimator (BUE) for μ. The precision of the location parameter estimation is directly influenced by skewness [4] and, hence, affects the confidence intervals and sample size [5,6].

Given that complex parametric distributions with several parameters are often considered [2], the information measures (entropies and/or divergences) play an important role in quantifying uncertainty provided by a random process about itself, and it is sufficient to study the reproduction of a marginal process through a noiseless system. One main application is related to the selection of models and detection of the number of clusters [7], or the interpretation of physical phenomena [8,9]. However, the use of entropies and/or divergences is widely considered to compare estimations [1]. For example, Ref. [10] considered the Kullback–Leibler (KL) divergence as a method to compare sample correlation matrices to an application in financial markets, assuming two multivariate normal densities. Using the estimated parameters based on maximum likelihood estimation, Ref. [11,12,13] considered the KL divergence for an asymptotic test to evaluate the data skewness and/or bimodality.

Given that precision was evaluated with confidence intervals in [5], the quantification of uncertainty for location parameter estimation under skew–normal settings motivated this study. The properties for MSE and (emphasizing) BUE, using classic theorems and properties of information theory are explored, which enable measuring the uncertainty of location parameter estimations based on differential entropy and Fisher information [1]. The Cramér–Rao inequality [14] linked Fisher information with the variance of an unbiased estimator, which is considered to find a lower bound for Fisher information. In addition, considering a stochastic representation [15] of a skew–normal random variable, the convexity property of Fisher information is also used to find an upper bound for Fisher information.

This paper is organized as follows: some properties and inferential aspects based on information theory are presented in Section 2. In Section 3, the computation and description of information–theoretic theorems related to location parameter estimation of skew–normal distribution are presented. In Section 4, some simulations illustrate the usefulness of the results. Final remarks conclude the paper in Section 5.

## 2. Information-Theoretic Aspects

In this section, some main theorems and properties of information theory are described. Specifically, these properties are based on differential entropy and Fisher information.

**Definition** **1.**
*Let X be a random variable with support in R and continuous probability density function (pdf), f(x;θ), which depends on parameter θ. The differential entropy of X [1] is defined by*

$$H\left(X\right)=-E\left[\log\;f(X;\theta )\right]=-\int_{R}f\left(x;\theta \right)\log\;f\left(x;\theta \right)dx,$$

*where notation $E\left[g\left(X\right)\right]=\int_{R}g\left(x\right)f\left(x;\theta \right)dx$ was used.*


Differential entropy depends only on the pdf of the random variable. In the following theorem, the scaling property of differential entropy is presented.

**Theorem** **1.**
*For any real constant a, the  differential entropy of aX Theorem 8.6.4 of [1] is given by*

H(aX)=H(X)+log|a|.



In particular, for two random variables, the following differential entropy bounds hold.

**Theorem** **2.**
*Let X and Y be two independent random variables. Suppose $Z\overset{D}{=}X+Y$, where “$\overset{D}{=}$” denotes equality in distribution, then*
*(i)* 

$$\frac{H(X)+H(Y)+\log2}{2}\leq H\left(Z\right)\leq H\left(X\right)+H\left(Y\right).$$

*(ii)* 
*For any constant ρ such that 0≤ρ≤1,*

$$\rho H\left(X\right)+\left(1-\rho \right)H\left(Y\right)\leq H\left(\sqrt{\rho }X+\sqrt{1-\rho }Y\right).$$




**Proof.** For part (i), consider first the general case for $X_{1},X_{2},\ldots ,X_{n}$ independent and identically distributed (i.i.d.) random variables see Equations (5) and (6) of [16], then
$$H\left(X_{1}\right)\leq H\left(\frac{X_{1}+X_{2}}{\sqrt{2}}\right)\leq \cdots \leq H\left(\frac{1}{\sqrt{n-1}}\sum_{i=1}^{n-1}X_{i}\right)\leq H\left(\frac{1}{\sqrt{n}}\sum_{i=1}^{n}X_{i}\right).$$
Considering the latter inequality for two variables, *X* and *Y*, and the scaling property of Theorem 1, we obtain 2H(X+Y)−log2≥H(X)+H(Y), yielding the left side of the inequality. For the right side, see [17]. The inequality of part (ii) is proved in Theorem 7 of [14].    □

The inequality of Theorem 2 (ii) is based on the convexity property, and allows obtaining a lower bound for differential entropy Theorem 8.6.5 of [1].

**Theorem** **3.**
*Let X be a random variable with zero mean and finite $\sigma^{2}$, then*

$$H\left(X\right)\leq \frac{1}{2}\log\left(2\pi e\sigma^{2}\right),$$

*and equality is achieved if, and only if, $X∼N(0,\sigma^{2})$.*


Theorem 3, also known as the maximum entropy principle, implies that Gaussian distribution maximizes the differential entropy over all distributions with the same variance. This theorem has several implications for information theory, mainly when the differential entropy of an unknown distribution is hard to obtain. Thus, this upper bound is a good alternative. Another consequence is the relationship between estimation error and differential entropy, which includes the Cramér–Rao bound as described next. First, the Fisher information for continuous densities needs to be defined as follows.

**Definition** **2.**
*Let X be a random variable with support in R and continuous density function f(x;θ), which depends on parameter θ, so $\int_{R}f\left(x;\theta \right)dx=1$. The Fisher information of X [1] is defined by*

(1)
$$J\left(X\right)=E\left[{\left\{\frac{\partial }{\partial x}\log f\left(x;\theta \right)\right\}}^{2}\right]=\int_{R}{\left(\frac{\partial }{\partial x}f\left(x;\theta \right)\right)}^{2}\frac{1}{f(x;\theta )}\;dx.$$



The Fisher information is a measure of the minimum error in estimating a parameter θ of a distribution. Classical definitions of Fisher information considered differentiation with respect to θ to define J(θ); however, by considering a parametric form as f(x−θ;θ), differentiation with respect to *x* is equivalent to differentiation with respect to θ as in Equation (Equation 1) [1]. The following inequality links Fisher information and variance.

**Theorem** **4.**
*Let $X={\left(X_{1},X_{2},\ldots ,X_{n}\right)}^{⊤}$ be a sample of n random variables drawn i.i.d. ∼f(x;θ), the mean-squared error of an unbiased estimator T(X) of parameter θ is lower bounded by the reciprocal of the Fisher information Theorem 11.10.1 of [1]:*

(2)
$$Var\left[T\left(X\right)\right]\geq \frac{1}{J(X)},$$

*where J(X) is defined in Equation (Equation 1) and, if the inequality is achieved, T(X) is efficient.*


Theorem 4, also known as the Cramér–Rao inequality, allows determining the best estimator of θ to obtain a lower bound for Fisher information. The Cramér–Rao inequality was first planned for any estimator T(X) (not necessarily unbiased) of θ in terms of mean-squared error, in this case
$$E\left[{\left\{T\left(X\right)-\theta \right\}}^{2}\right]\geq \frac{{\left(1+\frac{\partial }{\partial \theta }Bias\left(\theta \right)\right)}^{2}}{J(X)}+Bias{\left(\theta \right)}^{2},$$
where Bias(θ)=E[T(X)−θ]; see Equation (11).290 of [1]. Clearly, if T(X) is an unbiased estimator of θ, Theorem 4 is a particular case of the latter inequality. Inequality (Equation 2) was obtained through the Cauchy–Schwarz inequality on the variance of all unbiased estimators. The following inequality, also known as the Fisher information inequality, is based on the convexity property and is useful to obtain an upper bound for Fisher information.

**Theorem** **5.**
*For any two independent random variables X and Y, and any constant ρ such that 0≤ρ≤1, then*

$$J\left(\sqrt{\rho }X+\sqrt{1-\rho }Y\right)\leq \rho J\left(X\right)+\left(1-\rho \right)J\left(Y\right).$$



**Proof.** See proof of Theorem 13 in [14].    □

## 3. Location Parameter Estimation

The skew–normal distribution is an extension of the normal one, allowing for the presence of skewness.

**Definition** **3.**
*X is called a skew–normal random variable [15] and denoted as $X∼SN_{1}\left(\mu ,\sigma^{2},\lambda \right)$ if it has pdf*

$$f\left(x;\theta \right)=\frac{2}{\sigma }\phi \left(\frac{x-\mu }{\sigma }\right)\Phi \left[\lambda \left(\frac{x-\mu }{\sigma }\right)\right],\;x\in R,\;\theta =\left(\mu ,\sigma^{2},\lambda \right);$$

*with location μ∈R, scale $\sigma^{2}\in R$, and shape λ∈R parameters. In addition, ϕ(x) is the pdf of the standardized normal distribution with 0 mean and variance 1, denoted as N(0,1), and Φ(x) is the corresponding cumulative distribution function (cdf) of the standardized normal distribution.*


Random variable *X* is represented by the following stochastic representation:(3)$$X\overset{d}{\;=}\mu +\sigma (\delta |U_{0}|+\sqrt{1-\delta^{2}}U),$$
where $\delta =\frac{\lambda }{\sqrt{1+\lambda^{2}}}$, and $U_{0}$ and U∼N(0,1) are independently distributed; see Equation (2.14) of [15].

Additionally, *X* has a representation based on a link between differential entropy and Fisher information, due to *de Bruijn*’s identity. By matching the stochastic representation (Equation 3) with Equation (20) of [16], it is possible to assign $Y=\mu +\sigma \delta |U_{0}|$ with fixed δ. Then,
$$H\left(Y\right)=\frac{1}{2}\log\left(2\pi e\right)+\int_{-\infty }^{\infty }\lambda \left\{\left(1+\lambda^{2}\right)J\left(X\right)-1\right\}d\lambda ,$$
where an approximation for Fisher information J(X) appears in the proof of Proposition 5 below (with n=1 observation).

**Definition** **4.**
*X is called a multivariate skew–normal random vector [18] and denoted as $X∼SN_{n}\left(\mu ,\Sigma ,\lambda \right)$ if it has pdf*

$$f_{n}\left(x;\theta \right)=2\phi_{n}\left(x;\mu ,\Sigma \right)\Phi \left[\lambda^{⊤}\Sigma^{-1/2}\left(x-\mu \right)\right],\;x\in R^{n},\;\theta =\left(\mu ,\Sigma ,\lambda \right),$$

*with location vector $\mu \in R^{n}$, scale matrix $\Sigma \in R^{n\times n}$, and skewness vector $\lambda \in R^{n}$ parameters. In addition, $\phi_{n}\left(x;\mu ,\Sigma \right)$ is the n-dimensional normal pdf with location parameter **μ** and scale parameter ***Σ***.*


Let $X={\left(X_{1},\ldots ,X_{n}\right)}^{⊤}∼SN_{n}\left(\mu ,\Sigma ,\lambda \right)$, with $\mu =1_{n}\mu$, $\Sigma =\sigma^{2}I_{n}$, and $\lambda =1_{n}\lambda$, where $1_{n}={\left(1,\ldots ,1\right)}^{⊤}\in R^{n}$ and $I_{n}$ denotes the n×n-identity matrix. Following [3] and Corollary 2.2 of [5], the following properties hold.

**Property** **1.**
*$X_{i}∼SN_{1}\left(\mu ,\sigma^{2},\lambda_{*}\right)$, i=1,…,n, with $\lambda_{*}=\frac{\lambda }{\sqrt{1+\left(n-1\right)\lambda^{2}}}$.*


Property 1 indicates that $X_{1},\ldots ,X_{n}$ is a random sample with identically distributed but random variables not independent from a univariate skew–normal population with location μ, scale $\sigma^{2}$, and shape $\lambda_{*}$ parameters.

**Property** **2.**
*$E\left[X_{i}\right]=\mu +\sigma b\delta_{*}$ and $Var\left[X_{i}\right]=\sigma^{2}\left(1-nb^{2}\delta_{*}^{2}\right)$, with $b=\sqrt{\frac{2}{\pi }}$ and $\delta_{*}=\frac{\lambda }{1+n\lambda^{2}}$.*


**Property** **3.**
*$\bar{X}=\frac{1}{n}\sum_{i=1}^{n}X_{i}∼SN_{1}\left(\mu ,\frac{\sigma^{2}}{n},\sqrt{n}\lambda \right)$.*


**Property** **4.**
*$\left(\frac{n-1}{\sigma^{2}}\right)S^{2}∼\chi_{n-1}^{2}$ with $S^{2}=\frac{1}{n-1}\sum_{i=1}^{n}{\left(X_{i}-\bar{X}\right)}^{2}$, where $\chi_{n-1}^{2}$ denotes the chi-square distribution with n−1 degrees of freedom, and sample mean $\bar{X}$ and sample variance $S^{2}$ are independent.*


### 3.1. Least Square Estimator

Assuming that the coefficient of variation τ=|σ/μ| and shape parameter λ are known, Theorem 4.1 of [3] provides the least square estimator for μ and its variance, given by
(4)$$\begin{array}{ccc} T_{LSE}\left(X\right) & = & \omega \bar{X}, \end{array}$$
(5)$$\begin{array}{ccc} Var[T_{LSE}\left(X\right)] & = & \omega^{2}\left(1-n\delta_{n}^{2}\right)\frac{\sigma^{2}}{n},\\ \omega & = & \frac{n}{n+\tau \left(\frac{\tau +n\delta_{n}}{1+\delta_{n}\tau }\right)}, \end{array}$$
where $\delta_{n}=b\delta_{*}$, and $\delta_{*}$ is defined in Property 2. The least square estimator for μ was obtained by minimizing the MSE of $nc\bar{X}$ with respect to a constant *c*. The MSE of $T_{LSE}\left(X\right)$ is
(6)$$MSE\left[T_{LSE}\left(X\right)\right]=\left(\frac{\sigma^{2}}{n}+\mu^{2}+2\mu \delta_{n}\sigma \right)\omega^{2}-2\mu \omega \left(\mu +\sigma \delta_{n}\right)+\mu^{2}.$$

**Proposition** **1.**
*Let $X={\left(X_{1},\ldots ,X_{n}\right)}^{⊤}∼SN_{n}\left(\mu 1_{n},\sigma^{2}I_{n},1_{n}\lambda \right)$, with known τ and λ. Thus,*
*(i)* 

$$\begin{array}{ccc} H(T_{LSE}\left(X\right)) & = & \frac{1}{2}\log\left(2\pi e\frac{\sigma^{2}}{n}\omega^{2}\right)-H_{N}\left(\eta \right),\\ H_{N}\left(\eta \right) & = & E[\log\{2\Phi (\eta W)\}],\\ W & ∼ & SN_{1}\left(0,1,\eta \right),\\ \eta & = & \sigma \lambda . \end{array}$$

*(ii)* 

$$H\left(T_{LSE}\left(X\right)\right)\leq \frac{1}{2}\log\left(2\pi e\frac{\sigma^{2}}{n}\omega^{2}\left(1-n\delta_{n}^{2}\right)\right).$$




**Proof.** Part (i) follows straightforwardly from Theorem 1, Property 3, (Equation 4) and Proposition 2.1 of [19] (for the univariate case). Part (ii) is straightforward from Theorem 3 and (5).    □

Differential entropy of $T_{LSE}\left(X\right)$ corresponds to the difference of the normal differential entropy and a term called negentropy, $H_{N}\left(\eta \right)$, that depends on σ and λ parameters. Additionally, note that part (ii) yields the upper bound for $H_{N}\left(\eta \right)$ of part (i), $H_{N}\left(\eta \right)\leq \frac{1}{2}\log\left(1-n\delta_{n}^{2}\right)$.

As a particular case of Proposition 1, it is possible to obtain the differential entropy of sample mean $\bar{X}$ by choosing ω=1:(7)$$H\left(\bar{X}\right)=\frac{1}{2}\log\left(2\pi e\frac{\sigma^{2}}{n}\right)-E\left[\log\left\{2\Phi \left(\eta W\right)\right\}\right];$$
its respective upper bound
$$H\left(\bar{X}\right)\leq \frac{1}{2}\log\left(2\pi e\frac{\sigma^{2}}{n}\left(1-n\delta_{n}^{2}\right)\right);$$
and, from Equation (Equation 6), its respective MSE
(8)$$MSE\left[\bar{X}\right]=\frac{\sigma^{2}}{n}.$$

### 3.2. Best Unbiased Estimator

Assuming that the coefficient of variation τ=|σ/μ| and shape parameter λ are known, Theorem 5.1 of [3] provides the best unbiased estimator (BUE) for μ, given by
(9)$$\begin{array}{ccc} T_{BUE}\left(X\right) & = & \left(1-\alpha \right)d_{1}\left(X\right)+\alpha d_{2}\left(X\right),\\ d_{1}\left(X\right) & = & \frac{\bar{X}}{1+\delta_{n}\tau },\\ d_{2}\left(X\right) & = & c_{n}\sqrt{n-1}S, \end{array}$$
(10)$$\begin{array}{ccc} c_{n} & = & \frac{1}{\sqrt{2\tau^{2}}}\frac{\Gamma \left(\frac{n-1}{2}\right)}{\Gamma \left(\frac{n}{2}\right)}, \end{array}$$
(11)$$\begin{array}{ccc} \alpha & = & \frac{1}{\left(1+\delta_{n}\tau \right){\left[\left(n-1\right)c_{n}\right]}^{2}}, \end{array}$$
where Γ(x) denotes the usual gamma function and *S* is defined in Property 4.

**Remark** **1.**
*Equation (10) can be approximated using an asymptotic expression for the gamma function given by $\Gamma \left(x+a\right)\approx \sqrt{2\pi }x^{x+a-1/2}e^{-x}$, a<∞, as |x|→∞ [19]. Then,*

(12)
$$c_{n}\approx \frac{1}{\sqrt{n\tau^{2}}},$$

*as n→∞. Since the exact form (10) can be undefined for large samples (n>200), approximation (1) is very useful for these cases. Note that from (11) and (1), $\delta_{n},\alpha \to 0$ as n→∞, which implies that estimator $T_{BUE}\left(X\right)$ is only influenced by $d_{1}\left(X\right)$ for large samples.*


From Properties 3 and 4, Ref. [3] also proved that
(13)$$\begin{array}{ccc} d_{1}\left(X\right) & ∼ & SN_{1}\left(\frac{\mu }{1+\delta_{n}\tau },\frac{\sigma^{2}}{n{\left(1+\delta_{n}\tau \right)}^{2}},\sqrt{n}\lambda \right), \end{array}$$
(14)$$\begin{array}{ccc} Var[d_{1}\left(X\right)] & = & \frac{{\left(\mu \tau \right)}^{2}\left(1-n\delta_{n}^{2}\right)}{n{\left(1+\delta_{n}\tau \right)}^{2}}, \end{array}$$
(15)$$\begin{array}{ccc} Var[d_{2}\left(X\right)] & = & 2{\left(\mu \tau \right)}^{2}\left(1-\frac{1}{2\left(n-1\right){\left(\tau c_{n}\right)}^{2}}\right). \end{array}$$

Given that $Cov(d_{1},d_{2})=0$, from Equations (Equation 9), (14) and (15), we obtain
(16)$$\begin{array}{ccc} Var[T_{BUE}\left(X\right)] & = & {\left(1-\alpha \right)}^{2}Var\left[d_{1}\left(X\right)\right]+\alpha^{2}Var\left[d_{2}\left(X\right)\right]. \end{array}$$

The following proposition provides two upper bounds of differential entropy for $T_{BUE}\left(X\right)$ based on Theorem 3.

**Proposition** **2.**
*Let $X={\left(X_{1},\ldots ,X_{n}\right)}^{⊤}∼SN_{n}\left(\mu 1_{n},\sigma^{2}I_{n},1_{n}\lambda \right)$, with known τ and λ. Thus,*
*(i)* 

$$H\left(T_{BUE}\left(X\right)\right)\leq \frac{1}{2}\log\left(\frac{4\pi e}{n}{\left(\frac{\sigma^{2}\alpha \left(1-\alpha \right)}{1+\delta_{n}\tau }\right)}^{2}\left(1-\frac{1}{2\left(n-1\right){\left(\tau c_{n}\right)}^{2}}\right)\right),$$

*(ii)* 

$$H\left(T_{BUE}\left(X\right)\right)\leq \frac{1}{2}\log\left(2\pi eVar[T_{BUE}\left(X\right)]\}\right).$$




**Proof.** From Theorem 3 and Equations (14) and (15), the differential entropies of $d_{1}\left(X\right)$ and $d_{2}\left(X\right)$ are, respectively, upper bounded by
(17)$$\begin{array}{ccc} H(d_{1}\left(X\right)) & \leq & \frac{1}{2}\log\left(2\pi e\frac{\sigma^{2}}{n{\left(1+\delta_{n}\tau \right)}^{2}}\right), \end{array}$$
(18)$$\begin{array}{ccc} H(d_{2}\left(X\right)) & \leq & \frac{1}{2}\log\left(2\pi e2\sigma^{2}\left(1-\frac{1}{2\left(n-1\right){\left(\tau c_{n}\right)}^{2}}\right)\right). \end{array}$$Considering the right side on the inequality of Theorem 2(i), with Z=X+Y, $X=\left(1-\alpha \right)d_{1}\left(X\right)$ and $Y=\alpha d_{2}\left(X\right)$ (thus Cov(X,Y)=0), we obtain
$$H\left(T_{BUE}\left(X\right)\right)\leq H\left(\left(1-\alpha \right)d_{1}\left(X\right)\right)+H\left(\alpha d_{2}\left(X\right)\right)=H\left(d_{1}\left(X\right)\right)+H\left(d_{2}\left(X\right)\right)+\log\left|\left(1-\alpha \right)\alpha \right|,$$
where Theorem 1 is applied later. Then, Equations (Equation 17) and (18) yield part (i). On the other hand, by considering directly Theorem 3 on $T_{BUE}\left(X\right)$, Equation (Equation 16) implies part (ii).    □

The following proposition provides two lower bounds of differential entropy for $T_{BUE}\left(X\right)$.

**Proposition** **3.**
*Let $X={\left(X_{1},\ldots ,X_{n}\right)}^{⊤}∼SN_{n}\left(\mu 1_{n},\sigma^{2}I_{n},1_{n}\lambda \right)$, with known τ (0<τ<1) and λ. Thus*
*(i)* 

$$\frac{H\left(d_{1}\left(X\right)\right)+H\left(d_{2}\left(X\right)\right)+\log\left(2\alpha^{2}{\left(1-\alpha \right)}^{2}\right)}{2}\leq H\left(T_{BUE}\left(X\right)\right),$$

*(ii)* 

$$\left(1-\alpha^{2}\right)H\left(d_{1}\left(X\right)\right)+\alpha^{2}H\left(d_{2}\left(X\right)\right)\leq H\left(T_{BUE}\left(X\right)\right);$$


*where*

$$\begin{array}{ccc} H(d_{1}\left(X\right)) & = & \frac{1}{2}\log\left(\frac{2\pi e\sigma^{2}}{n{\left(1+\delta_{n}\tau \right)}^{2}}\right)-H_{N}\left(\eta_{1}\right),\\ H_{N}\left(\eta_{1}\right) & = & E[\log\left\{2\Phi \left(\eta_{1}W_{1}\right)\right\}],\\ W_{1} & ∼ & SN_{1}\left(0,1,\eta_{1}\right),\\ \eta_{1} & = & |\frac{\lambda \sigma }{1+\delta_{n}\tau }|; \end{array}$$

*and*

$$\begin{array}{ccc} H(d_{2}\left(X\right)) & = & \log\left(\frac{\left|\alpha \right|{\left[\Gamma \left(\frac{n-1}{2}\right)\right]}^{n}}{2{\left[\tau \Gamma \left(\frac{n}{2}\right)\right]}^{n-1}}\right)-\frac{\left(n-2\right)\Gamma \left(\frac{n-1}{2}\right)}{4{\left(2c_{n}^{2}\right)}^{-\left(\frac{n-1}{2}\right)}}\left(\psi \left(\frac{n-1}{2}\right)+\log\left(2c_{n}^{2}\right)\right)\\ & & +\;\frac{\Gamma (n+1)}{c_{n}^{2}}\frac{{\left[\Gamma \left(\frac{n-1}{2}\right)\right]}^{n+2}}{{\left[\tau \Gamma \left(\frac{n}{2}\right)\right]}^{n+3}}. \end{array}$$



**Proof.** Differential entropy of $d_{1}\left(X\right)$ is straightforward from evaluating (Equation 13) on Proposition 2.1 of [19] (for the univariate case). Given that distribution of $d_{2}\left(X\right)$ is unknown, Ref. [3] provided its pdf
(19)$$\begin{array}{c} f_{d_{2}}\left(x;\mu ,\sigma \right)=2\frac{{\left[\tau \Gamma \left(\frac{n}{2}\right)\right]}^{n-1}}{{\left[\Gamma \left(\frac{n-1}{2}\right)\right]}^{n}}x^{n-2}e^{-\frac{x^{2}}{2c_{n}^{2}}}. \end{array}$$Through Equations (Equation 19) and (3.381.4) of [20], the moments of $d_{2}$ are given by
(20)$$\begin{array}{c} E_{d_{2}}\left[X^{m}\right]=2\Gamma \left(m+n-1\right)\frac{{\left[\Gamma \left(\frac{n-1}{2}\right)\right]}^{2m+n-2}}{{\left[\tau \Gamma \left(\frac{n}{2}\right)\right]}^{2m+n-1}},\;m=0,1,\ldots ; \end{array}$$
and, using Equation (4.352.1) of [20], the moment of logx is
(21)$$\begin{array}{ccc} E_{d_{2}}\left[\log X\right] & = & \int_{0}^{\infty }f_{d_{2}}\left(x;\mu ,\sigma \right)\log xdx,\\ & = & 2\frac{{\left[\tau \Gamma \left(\frac{n}{2}\right)\right]}^{n-1}}{{\left[\Gamma \left(\frac{n-1}{2}\right)\right]}^{n}}\int_{0}^{\infty }x^{n-2}e^{-\frac{x^{2}}{2c_{n}^{2}}}\log xdx,\\ & = & \frac{{\left(2c_{n}^{2}\right)}^{\frac{n-1}{2}}\Gamma \left(\frac{n-1}{2}\right)}{4}\left(\psi \left(\frac{n-1}{2}\right)+\log\left(2c_{n}^{2}\right)\right), \end{array}$$
where $\psi \left(x\right)=\frac{d}{dx}\log\left\{\Gamma \left(x\right)\right\}$ is the digamma function. Therefore, by definition (1), the differential entropy of $d_{2}\left(X\right)$ is computed as
$$\begin{array}{ccc} H(d_{2}\left(X\right)) & = & -\int_{0}^{\infty }f_{d_{2}}\left(x;\mu ,\sigma \right)\log\;f_{d_{2}}\left(x;\mu ,\sigma \right)dx,\\ & = & -\log\left(\frac{2{\left[\tau \Gamma \left(\frac{n}{2}\right)\right]}^{n-1}}{\left|\alpha \right|{\left[\Gamma \left(\frac{n-1}{2}\right)\right]}^{n}}\right)-\left(n-2\right)\underset{E_{d_{2}}\left[\log X\right]}{\underset{︸}{\int_{0}^{\infty }f_{d_{2}}\left(x;\mu ,\sigma \right)\log xdx}}+\frac{1}{2c_{n}^{2}}\underset{E_{d_{2}}\left[X^{2}\right]}{\underset{︸}{\int_{0}^{\infty }f_{d_{2}}\left(x;\mu ,\sigma \right)x^{2}dx}}. \end{array}$$Thus, Equations (Equation 20) and (Equation 21) are evaluated in the latter expression to obtain $H(d_{2}\left(X\right))$. By assuming Z=X+Y in Theorem 2(i) (Cov(X,Y)=0), with $X=\left(1-\alpha \right)d_{1}\left(X\right)$ and $Y=\alpha d_{2}\left(X\right)$, the inequality of part (i) is obtained.By assuming Z=X+Y in Theorem 2(ii), with $X=d_{2}\left(X\right)$, $Y=d_{1}\left(X\right)$ (thus Cov(X,Y)=0), and $\rho =\alpha^{2}$, and since $d_{1}\left(X\right)$ and $d_{2}\left(X\right)$ are two unbiased estimators of μ [3], the inequality of part (ii) is obtained.    □

The following proposition provides a lower bound for Fisher information of parameter μ based on $T_{BUE}\left(X\right)$.

**Proposition** **4.**
*Let $X={\left(X_{1},\ldots ,X_{n}\right)}^{⊤}∼SN_{n}\left(\mu 1_{n},\sigma^{2}I_{n},1_{n}\lambda \right)$, with known τ and λ. Thus,*

$$\begin{array}{c} J\left(\mu \right)\geq {\left[{\left(1-\alpha \right)}^{2}\frac{\sigma^{2}\left(1-n\delta_{n}^{2}\right)}{n{\left(1+\delta_{n}\tau \right)}^{2}}+\alpha^{2}2\sigma^{2}\left(1-\frac{1}{2\left(n-1\right){\left(\tau c_{n}\right)}^{2}}\right)\right]}^{-1}. \end{array}$$



**Proof.** Considering that $T_{BUE}\left(X\right)$ is an unbiased estimator of μ, from the Crámer–Rao inequality of Theorem 4 and Equations (14)–(Equation 16), we obtain
$$J\left(\mu \right)\geq \frac{1}{{\left(1-\alpha \right)}^{2}Var\left[d_{1}\left(X\right)\right]+\alpha^{2}Var\left[d_{2}\left(X\right)\right]},$$
yielding the result.    □

The following Proposition provides an upper bound of Fisher information for parameter μ based on $T_{BUE}\left(X\right)$ and the convexity property.

**Proposition** **5.**
*Let $X={\left(X_{1},\ldots ,X_{n}\right)}^{⊤}∼SN_{n}\left(\mu 1_{n},\sigma^{2}I_{n},1_{n}\lambda \right)$, with known τ (0<τ<1) and λ. Thus*

$$J\left(\mu \right)\leq \left(1-\alpha^{2}\right)J\left(d_{1}\left(X\right)\right)+\alpha^{2}J\left(d_{2}\left(X\right)\right),$$

*where*

$$\begin{array}{ccc} J(d_{1}\left(X\right)) & \approx & 1+\frac{n{\left(b\lambda \right)}^{2}}{\sqrt{1+2nb^{4}\lambda^{2}}},\\ J(d_{2}\left(X\right)) & = & \frac{2\left(n-2\right)\left(n^{2}-2n-2\right)\Gamma \left(n-2\right)}{{\left(\frac{1}{2c_{n}^{2}}\right)}^{n-3}}\frac{{\left[\tau \Gamma \left(\frac{n}{2}\right)\right]}^{n-1}}{{\left[\Gamma \left(\frac{n-1}{2}\right)\right]}^{n}}+\frac{1}{c_{n}^{4}}, \end{array}$$



**Proof.** By assuming Z=X+Y in Theorem 5, with $X=d_{2}\left(X\right)$, $Y=d_{1}\left(X\right)$ (thus Cov(X,Y)=0), and $\rho =\alpha^{2}$, and since $d_{1}\left(X\right)$ and $d_{2}\left(X\right)$ are two unbiased estimators of μ [3], we obtain $J\left(\mu \right)\leq \alpha^{2}J\left(d_{2}\left(X\right)\right)+\left(1-\alpha^{2}\right)J\left(d_{1}\left(X\right)\right)$. Note that condition 0<τ<1 ensures that $0\leq \alpha^{2}\leq 1$.For $J(d_{1}\left(X\right))$, the steps of Section 3.2 of [9] were considered. By  Equations (Equation 1) and (Equation 13), and the change of variable $z=\left(x-\mu^{*}\right)/\sigma^{*}$, with $\mu^{*}=\mu /\left(1+\delta_{n}\tau \right)$, $\sigma^{*}=\sigma /\sqrt{n{\left(1+\delta_{n}\tau \right)}^{2}}$ and $\lambda^{*}=\sqrt{n}\lambda$, $J(d_{1}\left(X\right))$ can be computed as
(22)$$\begin{array}{ccc} J(d_{1}\left(X\right)) & = & \int_{-\infty }^{\infty }{\left(\frac{\partial }{\partial x}f\left(x;\theta \right)\right)}^{2}\frac{1}{f(x;\theta )}\;dx\\ & = & \int_{-\infty }^{\infty }f\left(z;\lambda^{*}\right){\left[-z+\lambda^{*}\zeta \left(\lambda^{*}z\right)\right]}^{2}dz\\ & = & \int_{-\infty }^{\infty }z^{2}f\left(z;\lambda^{*}\right)dz-2\lambda^{*}\int_{-\infty }^{\infty }z\zeta \left(\lambda^{*}z\right)f\left(z;\lambda^{*}\right)dz+{\left[\lambda^{*}\right]}^{2}\int_{-\infty }^{\infty }\zeta {\left(\lambda^{*}z\right)}^{2}f\left(z;\lambda^{*}\right)dz\\ & = & \int_{-\infty }^{\infty }z^{2}f\left(z;\lambda^{*}\right)dz-4\lambda^{*}\int_{-\infty }^{\infty }z\phi \left(\lambda^{*}z\right)\phi \left(z\right)dz+2{\left[\lambda^{*}\right]}^{2}\int_{-\infty }^{\infty }\phi \left(z\right)\frac{\phi {\left(\lambda^{*}z\right)}^{2}}{\Phi (\lambda^{*}z)}dz, \end{array}$$
where ζ(x)=ϕ(x)/Φ(x) is the *zeta* function. From Equation (Equation 22), the first and second terms are the second moment of a standardized skew–normal random variable ($E[Z^{2}]=1$) and the first moment of a standardized normal random variable (E[R]=0, R∼N(0,1)), respectively. The third term is
$$\begin{array}{ccc} \int_{-\infty }^{\infty }\phi \left(z\right)\frac{\phi {\left(\lambda^{*}z\right)}^{2}}{\Phi (\lambda^{*}z)}dz & = & \int_{0}^{\infty }\phi \left(z\right)\frac{\phi {\left(\lambda^{*}z\right)}^{2}}{\Phi (\lambda^{*}z)}dz+\int_{0}^{\infty }\phi \left(z\right)\frac{\phi {\left(\lambda^{*}z\right)}^{2}}{1-\Phi (\lambda^{*}z)}dz\\ & = & \int_{0}^{\infty }\phi \left(z\right)\frac{\phi {\left(\lambda^{*}z\right)}^{2}}{\Phi \left(\lambda^{*}z\right)\left[1-\Phi \left(\lambda^{*}z\right)\right]}dz \end{array}$$The following approximation of normal densities (see p. 83 of [15]),
$$\frac{\phi (y)}{\sqrt{\Phi (y)[1-\Phi (y)]}}\approx \pi b^{2}\phi \left(b^{2}y\right),\;\forall y\in R,$$
and some basic algebraic operations of normal densities are useful to approximate the third term of Equation (Equation 22) as
$$\begin{array}{ccc} \int_{0}^{\infty }\phi \left(z\right)\frac{\phi {\left(\lambda^{*}z\right)}^{2}}{\Phi \left(\lambda^{*}z\right)\left[1-\Phi \left(\lambda^{*}z\right)\right]}dz & \approx & \pi^{2}b^{4}\int_{0}^{\infty }\phi \left(z\right)\phi {\left(b^{2}\lambda^{*}z\right)}^{2}dz\\ & = & \frac{\pi^{2}b^{4}}{2\pi \sqrt{1+2b^{4}{\left[\lambda^{*}\right]}^{2}}}\int_{0}^{\infty }\phi \left(z;0,{\left\{1+2b^{4}{\left[\lambda^{*}\right]}^{2}\right\}}^{-1}\right)dz\\ & = & \frac{\pi b^{4}}{4\sqrt{1+2b^{4}{\left[\lambda^{*}\right]}^{2}}}. \end{array}$$Given that $\frac{\pi b^{4}}{2}=b^{2}$ and $\lambda^{*}=\sqrt{n}\lambda$, we obtain
$$\begin{array}{ccc} J(d_{1}\left(X\right)) & \approx & 1+\frac{n{\left(b\lambda \right)}^{2}}{\sqrt{1+2nb^{4}\lambda^{2}}}. \end{array}$$Using Equation (Equation 19), $J(d_{2}\left(X\right))$ is computed as
$$\begin{array}{ccc} J(d_{2}\left(X\right)) & = & \int_{0}^{\infty }{\left(\frac{\partial }{\partial x}f_{d_{2}}\left(x;\mu ,\sigma \right)\right)}^{2}\frac{1}{f_{d_{2}}\left(x;\mu ,\sigma \right)}\;dx\\ & = & \int_{0}^{\infty }f_{d_{2}}\left(x;\mu ,\sigma \right){\left[\left(n-2\right)x^{-1}-\frac{1}{c_{n}^{2}}\right]}^{2}dx\\ & = & {\left(n-2\right)}^{2}\int_{0}^{\infty }x^{-2}f_{d_{2}}\left(x;\mu ,\sigma \right)dx-2\frac{\left(n-2\right)}{c_{n}^{2}}\int_{0}^{\infty }x^{-1}f_{d_{2}}\left(x;\mu ,\sigma \right)dx+\frac{1}{c_{n}^{4}}\\ & = & 2{\left(n-2\right)}^{2}\frac{{\left[\tau \Gamma \left(\frac{n}{2}\right)\right]}^{n-1}}{{\left[\Gamma \left(\frac{n-1}{2}\right)\right]}^{n}}\underset{M_{1}}{\underset{︸}{\int_{0}^{\infty }x^{n-4}e^{-\frac{1}{2c_{n}^{2}}x^{2}}dx}}-\frac{4(n-2)}{c_{n}^{2}}\frac{{\left[\tau \Gamma \left(\frac{n}{2}\right)\right]}^{n-1}}{{\left[\Gamma \left(\frac{n-1}{2}\right)\right]}^{n}}\underset{M_{2}}{\underset{︸}{\int_{0}^{\infty }x^{n-3}e^{-\frac{1}{2c_{n}^{2}}x^{2}}dx}}+\frac{1}{c_{n}^{4}}\\ & = & \frac{2\left(n-2\right)\left(n^{2}-2n-2\right)\Gamma \left(n-2\right)}{{\left(\frac{1}{2c_{n}^{2}}\right)}^{n-3}}\frac{{\left[\tau \Gamma \left(\frac{n}{2}\right)\right]}^{n-1}}{{\left[\Gamma \left(\frac{n-1}{2}\right)\right]}^{n}}+\frac{1}{c_{n}^{4}}, \end{array}$$
where Equation (3.381.4) of [20] is applied to solve integrals $M_{1}$ and $M_{2}$.    □

**Remark** **2.**
*Considering the same argument as in Remark (1), it can be noted that inequalities of Propositions 4 and 5 are only affected by $J(d_{1}\left(X\right))$, i.e., for large samples, we obtain*

$$\frac{1}{Var[d_{1}\left(X\right)]}\leq J\left(\mu \right)\leq J\left(d_{1}\left(X\right)\right).$$



## 4. Simulations

All location parameter estimators, variances, Fisher information and differential entropies were calculated with R software [21]. Samples based on skew–normal random variables were drawn based on stochastic representation (Equation 3) and with the rsn function of sn package. All R codes used in this paper are available upon request from the corresponding author.

In general, τ takes a value between 0 and 1. If τ is close to 0, the sample has low variability, and if it is close to 1, the sample has high variability and mean loss reliability. For example, if τ>0.3, the mean is less representative of the sample. Sometimes, if μ is close to zero, τ takes high values (high variability) and could exceed unity. Therefore, for illustrative purposes, in all simulations, a coefficient of variation set of τ=0.1,…,1 is considered. Additionally considered are positive asymmetry parameters λ=0.1,…,5, sample sizes n=10 and 250, and theoretical location parameters μ=0.1, 0.5 and 1. For the computation of information measures, σ is replaced by τ|μ| and location parameters μ are evaluated by their respective estimators.

The MSE of $T_{LSE}\left(X\right)$ is given in Equation (Equation 6), and MSE of $T_{BUE}\left(X\right)$ is the variance of the (unbiased) estimator (see Equation (Equation 16)). Without loss of generality, τ=1 is considered in Figure 1 because the same pattern is repeated for values of τ between 0 and 1. Comparing the MSE of both estimators, Figure 1 shows for all cases that differences between MSEs tend to increase for large values of μ, and MSEs turn around a specific value when the sample size increases. Moreover, MSEs of the unbiased estimator are less than those obtained by LSE, i.e., BUE dominates LSE Equation (11.263) of [1]. Therefore, the analysis focuses on BUE in the next section.

Behavior of differential entropy bounds given in Propositions 2 and 3 is illustrated in Figure 2 as 3D plots. Without loss of generality, τ∈(0,1] is considered in Figure 2 because the same pattern is repeated for values of τ>1, i.e., entropies keep increasing. The upper bound corresponds to the minimum value between bounds given in Proposition 2(i) and (ii), which is the one given in (ii). Thus, the upper bound of $H(T_{BUE}\left(X\right))$ is determined by the variance (or MSE) of the estimator. In contrast, the lower bounds correspond to the maximum value between bounds given in Proposition 3 (i) and (ii) [17].

Sample sizes (n=250) imply that α≈0 and lower bounds only depend on $H(d_{1}\left(X\right))$. For small sample sizes (n=10), α could be an intermediate value of the interval (0,1), thus, lower bounds depend on $H(d_{1}\left(X\right))$ and $H(d_{2}\left(X\right))$. For n=10, the surfaces are rough, given the randomness of bounds produced by the small sample. When λ≈0 (symmetry condition), the bounds decay to negative values. This is analogous to considering the skew–normal density as a non-stationary process [15], when λ is near zero, so the Hurst exponent decreases abruptly [8]. On the other hand, for n=250, surfaces are soft and bounds increase slightly for large λ. For all cases, information increases when τ tends to 1 because it produces more variability in samples.

For practical purposes, the average between bounds can be considered to provide an approximation of differential entropy [7] in similar form to average lengths of the confidence interval [3]. Given that all lower bounds of differential entropy depend on the entropy of $d_{1}\left(X\right)$, which depends on variance and sample size, they could take negative values and tend to zero when τ tends to 1. Therefore, the difference between lower and upper bounds could increase and turn out an inadequate approximation if the lower bound is negative. For the latter reason, the Fisher information considers only positive values, as studied next.

The Fisher information bounds given in Propositions 4 and 5 are illustrated in Figure 3 as 3D and 2D plots, respectively. As in the differential entropy case, and without loss of generality, τ∈(0,1] is considered in Figure 3 because the same pattern is repeated for values of τ>1, i.e., entropies keep decreasing. Following the Cramér–Rao theorem, the variance of BUE corresponds to the reciprocal of the Fisher information. In contrast, the lower bound corresponds to a combination of the Fisher information of $d_{1}\left(X\right)$ and $d_{2}\left(X\right)$.

As in the differential entropy case, large sample sizes (n=250) imply that α≈0 and lower bounds only depend on $J(d_{1}\left(X\right))$, as mentioned in Remark 2. For small sample sizes (n=10), α could be an intermediate value of the interval (0,1), thus lower bounds depend on $J(d_{1}\left(X\right))$ and $J(d_{2}\left(X\right))$. When τ≈0 (low variability condition), the lower bounds take the highest values. This reciprocal relationship is determined by the Cramér–Rao theorem: more variability, less Fisher information. In addition, the 2D plot shows that the smallest upper bounds of $J(T_{BUE}\left(X\right))$ are produced when λ≈0 [9]. Given that upper bounds do not depend on τ and μ because the skew–normal densities are standardized, these measures are illustrated with respect to *n* and λ. In addition, when λ and *n* increase simultaneously, the upper bounds of Fisher information take the largest values.

## 5. Concluding Remarks

In this paper, some properties of the best unbiased estimator proposed by [3] were presented, using classic theorems of information theory, which provide a way to measure the uncertainty of location parameter estimations. Given that BUE dominates LSE, this paper focused on this estimator. Inequalities based on differential entropy and Fisher information allowed obtaining lower and upper bounds for these measures. Some simulations illustrated the behavior of differential entropy and Fisher information bounds.

Classical theorems of information theory considered the obtained additional properties of unbiased location parameter estimators. However, these theorems could be applied to other estimators, such as Bayesian [22] (as long as the prior pdf density is known), shrinkage [23], or bootstrap-based [24] ones. The assumption of the sample that came from a multivariate skew–normal distribution is too strong and not always applicable in the real world, so the properties revised here could be extended to more complex densities, for example, those that assess bimodality and heavy tails in data [7,11,13,19]. On the other hand, and given that Fisher information bounds under skew–normal settings were considered in this study, further work could focus on developing time-dependent Fisher information for skew–normal density [25], which could be applied to real data in survival analysis.

## Figures and Tables

**Figure 1 entropy-24-00399-f001:**
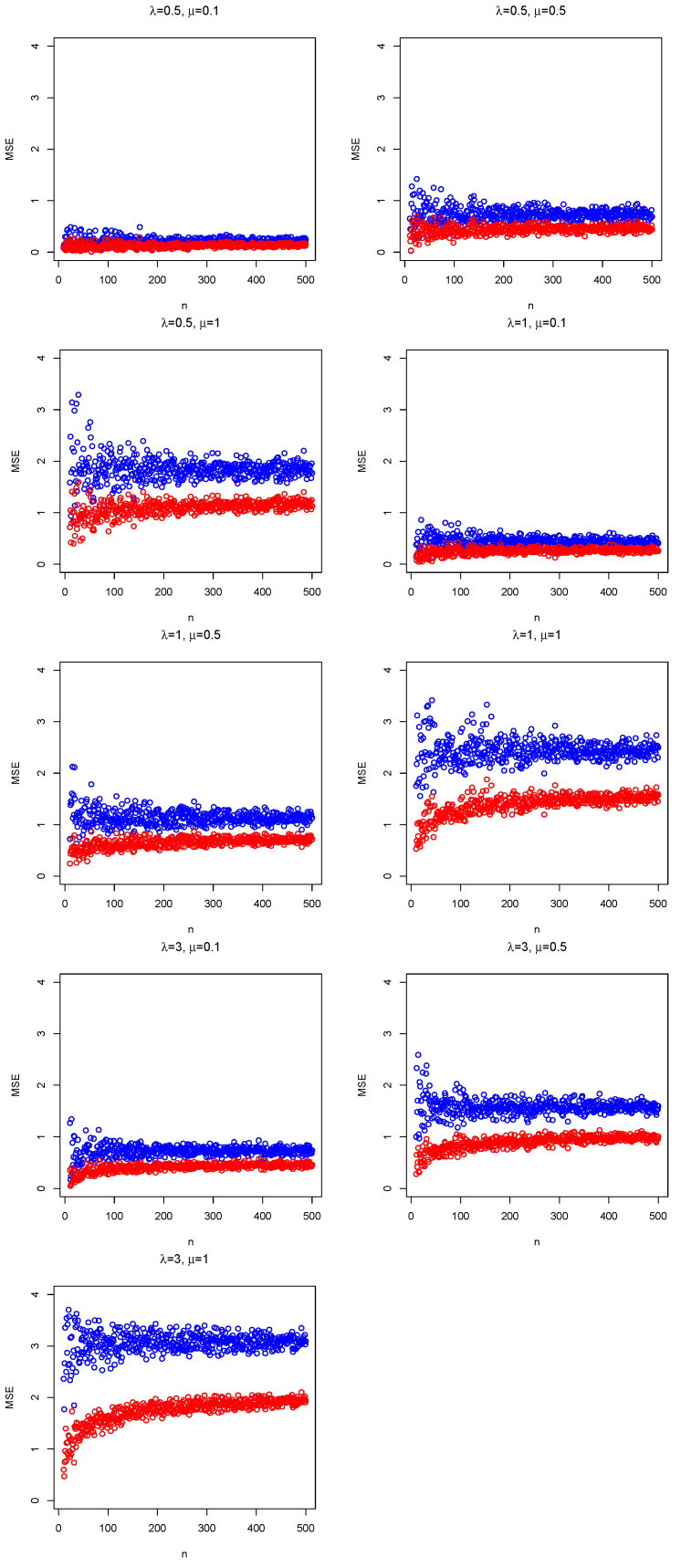
Mean square errors (MSE) for $T_{LSE}\left(X\right)$ [blue dots] and $T_{BUE}\left(X\right)$ [red dots] considering τ=1 and several skewness λ and location μ parameters in the simulations.

**Figure 2 entropy-24-00399-f002:**
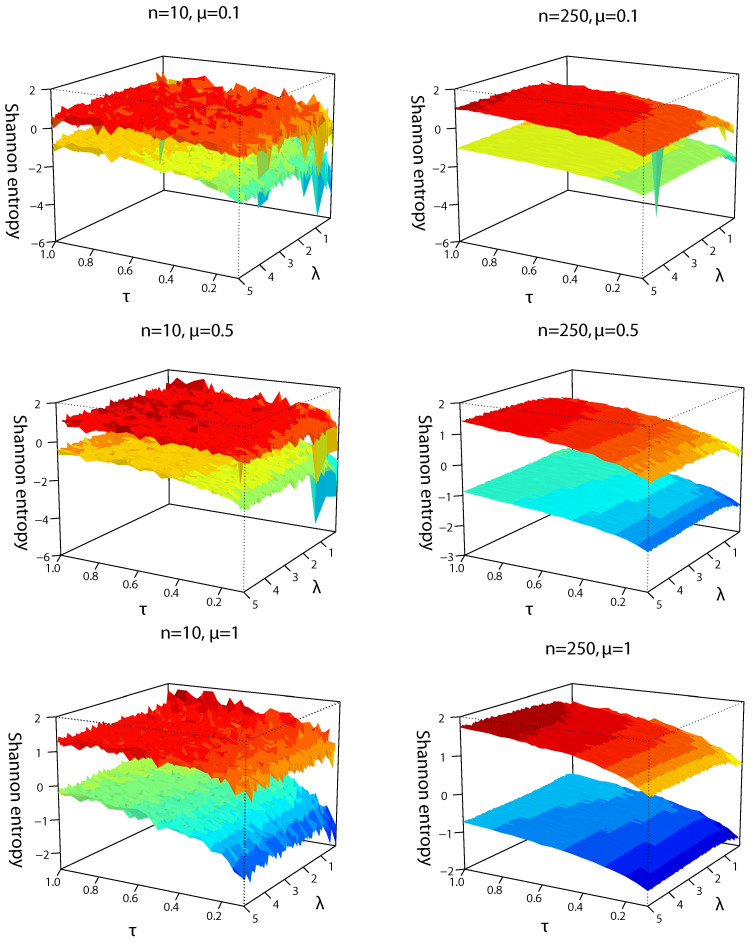
Differential entropy bounds for $T_{BUE}\left(X\right)$ considering n=10 and 250, μ=0.1, 0.5 and 1; and several skewness λ and coefficient of variation τ parameters in the simulations.

**Figure 3 entropy-24-00399-f003:**
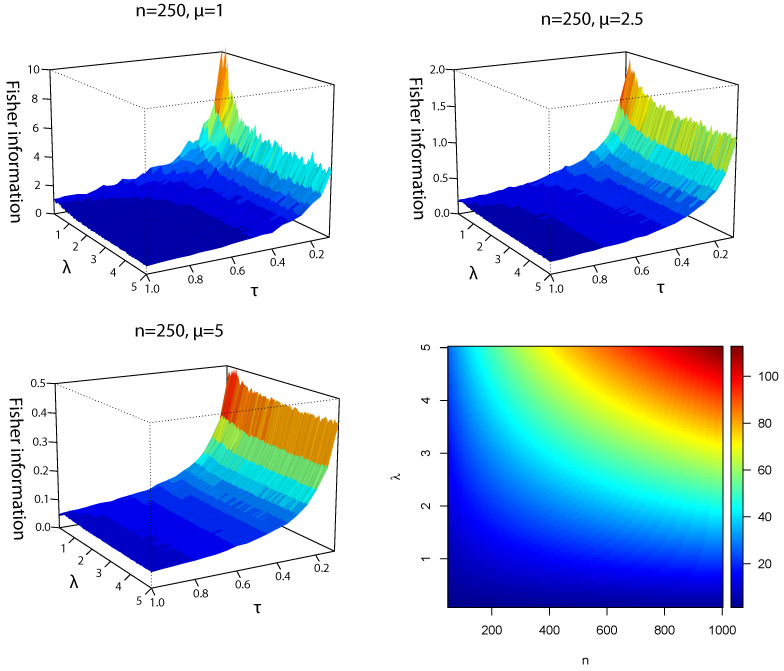
Fisher information lower bounds for $T_{BUE}\left(X\right)$ considering n=250, μ=1, 2.5 and 5; and several skewness λ and coefficient of variation τ parameters in the simulations. The fourth panel shows the upper bounds for $T_{BUE}\left(X\right)$ considering n=100,…,1000 and several skewness parameters λ.

## Data Availability

Not applicable.

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
