# Peer review of "Information–Theoretic Aspects of Location Parameter Estimation under Skew–Normal Settings"

_entropy, 2022, doi:10.3390/e24030399_

Round 1

Reviewer 1 Report

I do not have any specific comments. I suggest the publication of the paper as it is.

Author Response

Dear Reviewer:

I would like acknowledge this careful revision of our manuscript entropy-1617729 titled: "Information-theoretic aspects of location parameter estimation under skew-normal settings". I am grateful that this manuscript can be considered for publication.

Reviewer 2 Report

The paper is very well-written. Results and conclusions are properly described. I recommend the paper for publication in its present form.

Author Response

(The authors gave the same response as above.)

Reviewer 3 Report

There are some problems with determiners.

There are some problems with formulas that I have marked on the submitted version. 

A major peeve is not using standard notation for the covariance matrix. 

Author Response

Dear Reviewer:

I would like acknowledge this careful revision of our manuscript entropy-1617729 titled: "Information-theoretic aspects of location parameter estimation under skew-normal settings". I am grateful that this manuscript can be considered for publication after minor revision. We also thank the reviewer for all their valuable comments and constructive criticism. I have included (see below), a detailed point-by-point response to all the reviewer's comments and suggestions.

1. There are some problems with determiners.

R: All these determiners were fixed as suggested and changes marked in red.

2. There are some problems with formulas that I have marked on the submitted version.

R: All these problems were fixed as suggested and changes marked in red.

3. A major peeve is not using standard notation for the covariance matrix. In Definition 4: I do not like this notation nor its name. Isn’t this the covariance matrix, typically called $\Sigma$? What change a typical pattern of understanding this pdf?

R: This is the scale matrix (not necessarily covariance matrix). However, following your suggestion its was changed to Sigma. Alos, this is another pdf (n-dimensional version), and is different to univariate version.

4. L187: This is hardly a high number of observations.

R: I am agree with referee that "High" and "large" words are subjective. Thus all these words (and small) were removed.

5. L278: name of first author is correct. Now, all references follow the style required by Entropy journal.